The potential mechanism of Huangqin for treatment of systemic lupus erythematosus based on network pharmacology, molecular docking and molecular dynamics simulation

Zheng Shuting 1
Yang Hui 2
Wu Jialing 1
Jin Ou 1 jinou@mail.sysu.edu.cn
Zhang Xi 1 zhangx238@mail.sysu.edu.cn
1 Department of Rheumatology and Immunology, Third Affiliated Hospital of Sun Yat-Sen University , Guangzhou, Guangdong , China
2 Department of Rheumatology and immunology, People’s Hospital of Ningxia Hui Autonomous Region , Yinchuan, Ningxia Hui Autonomous Region , China
Nakai Kenta
Electronic publication date: 2025 Jun 26
Publication date: 2025
Volume: 13
Electronic Location ID: e19536
Received 2024 Nov 7; Accepted 2025 May 7
Copyright: © 2025 Yang et al.
Copyright year: 2025
Copyright holder: Yang et al.
License: This is an open access article distributed under the terms of the Creative Commons Attribution License, which permits unrestricted use, distribution, reproduction and adaptation in any medium and for any purpose provided that it is properly attributed. For attribution, the original author(s), title, publication source (PeerJ) and either DOI or URL of the article must be cited.
License URL: https://creativecommons.org/licenses/by/4.0/

Keywords: Systemic lupus erythematosus, Network pharmacology, Scutellaria baicalensis, Chinese medicine, Molecular docking, molecular dynamics simulation

Funding: Scientific Research Project of Guangdong Provincial Bureau of Traditional Chinese Medicine 20231059 Guangzhou Science and Technology Bureau Young Doctor Program 2024A04J4705 This work was supported by the Scientific research project of Guangdong Provincial Bureau of Traditional Chinese Medicine (20231059) and Guangzhou Science and Technology Bureau Young Doctor program (2024A04J4705). The funders had no role in study design, data collection and analysis, decision to publish, or preparation of the manuscript.

==============================
Background

Systemic lupus erythematosus (SLE) is an autoimmune disease that cannot be cured at present. The primary treatment strategies for SLE include glucocorticoids, immunosuppressants, antimalarial drugs, and biologics. There is an urgent need for milder and more effective treatment methods. This study aims to explore the effective ingredients and targets of traditional Chinese medicine Scutellaria baicalensis (Huangqin) in the treatment of systemic lupus erythematosus based on network pharmacology, and further analyze the potential mechanisms of action.

Method

Employing the Traditional Chinese Medicine Systems Pharmacology Database and Analysis Platform (TCMSP) database to identify the active chemical components of Huangqin, searching for target genes related to SLE through GeneCards and the KEGG database, extracting the SLE expression gene profile from the GEO database to identify SLE-related targets, and identifying Huangqin-SLE interaction targets using Venny diagrams; Constructing protein interaction networks using the STRING database, identifying core targets with Cytoscape software, and conducting protein clustering analysis; Importing the common targets into the Database for Annotation, Visualization and Integrated Discovery (DAVID) for Gene Ontology (GO) and Kyoto Encyclopedia of Genes and Genomes (KEGG) enrichment analysis. Molecular docking was carried out by AutoDockTools, AutoDock Vina, and Discovery Studio software to verify the correlation between the main components of Huangqin and the core targets. Molecular dynamics simulations further confirmed the stability of the binding between components and the targets.

Results

Network pharmacology identified 36 active components and 512 drug targets in Huangqin, resulting in the identification of 1,439 SLE targets and 28 common targets. The primary active components are baicalin, acacetin, oroxylin a, neobaicalin, and stigmasterol. Key genes were IL6, CASP3, BCL2, ESR1, and FOS; GO enrichment analysis yielded 77 significant results, while KEGG enrichment analysis produced 64 significant results. The primary signaling pathways targeted by Huangqin in SLE include the HIF-1 signaling pathway, PI3K-AKt signaling pathway, MAPK signaling pathway, IL-17 signaling pathway. Results of molecular docking indicated good binding affinity of Huangqin to stigmastero, baicalein and acacetin. The dynamics simulation indicated that the four complexes possessed reliable structural stability and compactness.

Conclusion

Huangqin can treat SLE through its effective components such as baicalin and acacetin. The mechanism involves inhibiting inflammatory factors, affecting the expression and activation of immune cells, and regulating cell autophagy.

Introduction

Systemic lupus erythematosus (SLE) is a kind of autoimmune disease that predominantly affecting women of childbearing age. It is characterized by the loss of immune tolerance and the production of autoantibodies, particularly antinuclear antibodies. Clinically, SLE often presents with multi-system involvement, showing a wide range of manifestations that can be severely debilitating and life-threatening. Current treatment options for SLE include glucocorticoids, immunosuppressants, antimalarial drugs, and biologics, all of which can have various side effects and adverse reactions with long-term use. There is an urgent need for a gentle yet effective treatment method. In recent years, an increasing number of studies have found that traditional Chinese medicine can significantly control the disease activity of SLE patients and reduce the dosage of glucocorticoids used (Wang et al., 2021a). Traditional Chinese herbal formulas and single herbs have shown significant efficacy in alleviating SLE symptoms by regulating cellular autophagy (Tian et al., 2022), modulating T cell survival, intervening in the balance of Th17/Treg cells (Xia & Huanping, 2019; Fengjiao et al., 2023), and reducing the secretion of pro-inflammatory cytokines. Clinical studies have shown that combining Chinese herbal medicine with conventional treatments can improve patient symptoms and shorten treatment duration. Yan (2020) used a combination of cyclophosphamide and sophora japonica based on glucocorticoids for the treatment of SLE patients, and showed that the total dose of cyclophosphamide and the average daily dose of glucocorticoids in the drug combination group was significantly reduced compared to the group treated with hormones alone, and the SLEDAI score and urinary protein also significantly improved. Zhang et al. (2021) used the formula Fuzheng Jiedu Tang combined with methylprednisolone for treatment. The results showed that compared with the group treated with glucocorticoids alone, the combined treatment group exhibited a significant reduction in SLEDAI scores, serum IFN-γ levels, and bone marrow suppression rates.

Scutellaria baicalensis (Huangqin) is a perennial herb of the Lamiaceae family, commonly used in folk Chinese medicine in combination with various other medicinal materials for the treatment of systemic lupus erythematosus. Research and analysis of patent prescriptions for the treatment of SLE show that the high-frequency Chinese medicinal materials include Rehmannia glutinosa, Astragalus membranaceus, Glycyrrhiza uralensis, Moutan cortex, Angelica sinensis, Salvia miltiorrhiza, Lithospermum erythrorhizon, Paeonia lactiflora, Poria, and Scutellaria baicalensis (Min et al., 2022). Liu et al. (2017) added Artemisia annua and Scutellaria baicalensis to the Wen Dan Tang, forming the Haoqin Wendan Tang, which was used for two patients with systemic lupus erythematosus. The symptoms such as fever, rash, and tinnitus were relieved, and the dosage of hormones was gradually reduced. Studies have shown that baicalin mouthwash can effectively reduce the probability of oral fungal infections in SLE patients using glucocorticoids during the active phase (Chunman et al., 2016). Although knowing Huangqin is effective in treating SLE, the active components and the possible pathways are still unknown.

Recently, research has shown that Scutellaria baicalensis has antiviral, antibacterial, anti-inflammatory, anti-tumor, and antioxidant functions (Lai et al., 2003; Xiao & Hogger, 2013). Some studies have shown that Huangqin Qingre Jiebi capsule can affect rheumatoid arthritis through multiple signaling pathways such as FZD8-Wnt/β-catenin, and HIF-1α/VEGF (Zhou, 2023; Hongli, Chenyu & Qing, 2022). Wei et al. (2023) found that baicalin can reduce the activation of NLRP3 inflammasomes in fibroblast-like synoviocytes of rheumatoid arthritis by regulating the let-7i-3p-PI3K/Akt/NF-κB signaling axis. As the active component of Chinese medicine Huangqin, baicalin has also been found in recent years to have a unique effect in the field of anti-inflammatory, and studies have shown that baicalin has significant effects on liver damage repair (Shi et al., 2018), anti-pulmonary fibrosis (Huang et al., 2016), and ulcerative colitis (Liang et al., 2019). At present, the effect of baicalin on SLE still needs further exploration and experimental confirmation.

Network pharmacology and bioinformatics integrate the basic theories and research tools of medicine, biology, computer science, and other disciplines. It is an innovative method based on systems biology, including the construction and enrichment analysis of drug-target and drug-disease networks. This method aims to reveal the complexity of biological systems, drugs, and diseases and can analyze and predict the pharmacological action mechanisms of drugs (Wu et al., 2018). Molecular docking can predict the binding mode and binding free energy between proteins and ligands, further verifying the function and mechanism of action of drugs (Crampon et al., 2022). Molecular dynamics simulation can further verify the binding modes between these molecules and protein receptors, providing a more precise molecular-level understanding (Bai et al., 2023). By far, although we know that Huangqin is a powerful herb in the treatment of SLE, we still know little about the explicit molecular mechanism about how Huangqin treat SLE. In this study, network pharmacology, bioinformatics, and other means are used to systematically analyze the mechanism of action of baicalin in the treatment of systemic lupus erythematosus from the molecular level, which is instructive for the further development of effective components of baicalin for the treatment of SLE. The workflow of this study was depicted in Fig. 1.

Figure 1 The workflow.

Materials and Methods

Identification of active components and targets of Huangqin

The Traditional Chinese Medicine Systems Pharmacology Database and Analysis Platform (TCMSP, tcmsp-e.com) was utilized to retrieve information on the physicochemical properties and target action information of the active components of Scutellaria baicalensis (Ru et al., 2014). Criteria of effective ingredients including oral bioavailability (OB) and drug-likeness (DL). Based on the criteria provided in existing literature, the selection criteria were set to OB > 30%, DL > 0.18, and a total of 36 effective components and 512 targets were selected. All target information was standardized through the UniProt database (https://www.uniprot.org/) and a network of active components and target genes was constructed using Cytoscape 3.7.2.

Disease target screening and acquisition

The GeneCards database (https://www.genecards.org/) was used with the search query “[disorders](systemic lupus erythematosus)” to identify related genes, and the KEGG database (https://www.kegg.jp/) was searched for “Systemic lupus erythematosus” to supplement disease genes.

The GSE database (GSE51997) was utilized to obtain gene expression profiles from normal individuals and lupus patients (Kyogoku et al., 2013; Menssen et al., 2009). The dataset includes four healthy individuals and six lupus patients. Data was imported into R 4.3.1 and differential gene analysis was conducted using the limma R package with a set condition of P < 0.05, |logFc| > 1. The ggplot2 and ComplexHeatmap R packages were used to create volcano plots and gene heatmaps.

Construction and analysis of the PPI network

Add the intersection targets of Huangqin and SLE in the STRING database (https://cn.string-db.org/) (Szklarczyk et al., 2021), with “Homo Sapiens” set as the species, and all other parameters set to default, to obtain the Protein-Protein interaction (PPI) network. The network file was imported into Cytoscape v3.7.2 for processing, and the protein-protein interaction network diagram was redrawn. The MCC algorithm in the cytoHubba plugin of Cytoscape was used to identify the top five hub genes (Chin et al., 2014). The MCODE program in Cytoscape was used for protein clustering, and the cluster was further subjected to BP enrichment analysis.

GO enrichment analysis and KEGG enrichment analysis

The Database for Annotation, Visualization and Integrated Discovery (DAVID, https://david.ncifcrf.gov) is used to perform GO enrichment analysis and KEGG enrichment analysis on the key targets. GO analysis was conducted in three aspects: biological process (BP), molecular function (MF), and cellular component (CC). Results with an FDP ≤ 0.05 were selected, and visualization was performed using the “ggplot2” R software package. The top 50 KEGG pathways were categorized and summarized based on the six classifications in the KEGG PATHWAY database. Non-SLE-related pathways were excluded from the KEGG pathway enrichment analysis. A targeted pathway network was constructed using Cytoscape 3.7.2, the targeted areas and color tones in the network were arranged according to the Degree value.

Molecular docking validation

Molecular docking between the core components and the core targets was performed based on the above analysis. The 3D crystal structures of the target proteins were obtained in the Protein Data Bank (PDB) (Drew et al., 1981). Heir PDB format files were downloaded, dehydrated and hydrogenated in AutoDockTools (version 1.5.7), then selected as receptors and saved as pdbqt files. The mol2 files of the active components were obtained from the TCMSP database, hydrogenated in AutoDockTools, selected as ligands, and exported as pdbqt files. Molecular docking was performed using AutoDock Vina (version 1.2.3) (Eberhardt et al., 2021; Trott & Olson, 2010), and the affinity of each component to the target was obtained. Components and targets with good binding activity were selected based on affinity and visualized using Discovery Studio 2019.

Molecular dynamic simulation

Molecular dynamics (MD) simulations were performed using the Gromacs2022 program, with the Generalized Amber Force Field (GAFF) for small molecules, the AMBER14SB force field for proteins, and the TIP3P water model, with the system temperature set at 298 K and the simulation time at 100 nanoseconds. After the simulation was completed, the simulation trajectories were analyzed using VMD and PyMOL, and the g_mmpbsa program was used to perform the MMPBSA binding free energy analysis between the protein and the small molecule ligand.

Results

Effective components and targets of Huangqin

A total of 36 active components and 512 drug targets of Scutellaria baicalensis were identified through the TCMSP database (tcmsp-e.com). The top five components with the highest degree values analyzed by NetworkAnalyst in Cytoscape were: baicalin, acacetin, oroxylin a, neobaicalin, and stigmasterol (Table 1, Table S1). Hexagons represent the active components of Huangqin and rhombuses represent the targets of the active components. The degree value represents the number of edges connected to the node (Fig. 2).

Table 1 The 5 core ingredients of Huangqin.

Name	Degree	Betweenness	Closeness	
Baicalein	12	0.080790675	0.477477477	
Acacetin	8	0.03785432	0.445378151	
Oroxylin a	7	0.032583607	0.438016529	
Neobaicalein	6	0.013356842	0.430894309	
Stigmasterol	6	0.032105119	0.430894309	

Figure 2 Construction of the Huangqin–SLE–targets network.

Identification of targets related to Scutellaria baicalensis intervention in SLE

Using the GeneCards database (https://www.genecards.org/), 5,947 related genes were filtered out, and the top 1,493 were selected based on the Score value. The KEGG database (https://www.kegg.jp/) provided 23 disease genes for “Systemic lupus erythematosus.”

Based on the GSE database GSE51997 microarray dataset, which includes four healthy individuals and six lupus patients, a total of 160 up-regulated genes and 35 down-regulated genes were obtained (Table S2). Volcano maps and heat maps of DEGs were shown in Figs. 3A and 3B.

Figure 3 SLE target screening and acquisition.

(A) Volcano plot of differently expressed genes between SLE patients and normal people in GSE51997 dataset (B) heat map of differently expressed genes between SLE patients and normal people in GSE51997 dataset.

After merging the genes obtained from the three databases and removing duplicates, a total of 1,639 disease-related genes were obtained. A Venn diagram was used to identify the intersecting targets between the predicted Huangqin targets and SLE targets. The analysis identified 28 targets as potential candidates for the anti-SLE action of Huangqin (Fig. 4A).

Figure 4 Targets of SLE-Huangqin and PPI analysis.

(A) Venn diagram of targets of SLE with the targets of Huangqin active components. (B) The PPI network of intersection targets of Huangqin interfering with SLE intersection targets.

Construction and analysis of PPI network

A PPI network containing 28 nodes and 334 edges was constructed by adding the intersection targets to the STRING database (https://cn.string-db.org/) with “Homo Sapiens” specified as the species. The network file was imported into Cytoscape v3.7.2 for processing (Fig. 4B, Table S3). The darker the color of the nodes in the diagram, the greater the impact on SLE. Protein clustering of the targets was performed using the MCODE plugin in Cytoscape, and BP enrichment analysis was further conducted on the cluster, with the seed protein of the cluster being CYCS (Fig. 5A), mainly involved in the signal transduction of intracellular steroid hormone receptors, positive regulation of transcription, and response to estradiol (Fig. 5B). The top five hub genes identified by the MCC algorithm in the cytoHubba plugin of Cytoscape are IL6, CASP3, BCL2, ESR1, and FOS (Fig. 5C).

Figure 5 Results of PPI network analysis.

(A) The protein clusters obtained in the MCODE plug-in analysis. (B) GO-BP analysis of the protein clusters in (A). (C) The top five hub genes.

GO and KEGG enrichment analysis

Gene ontology (GO) analysis showed that the main enriched biological processes (BP) targeted by Scutellaria baicalensis in SLE include positive regulation of transcription by RNA polymerase II promoter, positive regulation of transcription, response to estradiol, response to external stimulus, mammary gland alveolar development, and response to hypoxia; the main cellular components (CC) include macromolecular complexes, chromatin, caspase complexes, nucleoplasm, and transcription factor complexes; the main molecular functions enriched include RNA polymerase II transcription factor activity, transcription coactivator binding, enzyme binding, and sequence-specific DNA binding (Fig. 6A, Table S4).

Figure 6 The results of gene enrichment analysis-1.

(A) GO enrichment analysis. (B) Bar graph of KEGG enrichment analysis (top 50 results).

The main Kyoto Encyclopedia of Genes and Genomes (KEGG) pathways involved in SLE by Scutellaria baicalensis include the Pathways in cancer, Lipid and atherosclerosis, Kaposi sacroma, measles, and so on (Fig. 6B, Table S5).

Figures 7A and 7B showed the top 50 enriched pathways, which can be divided into four categories: cellular processes (cell growth and death, cell communication), organism systems (endocrine system, immune system, nervous system), human diseases (tumors, cardiovascular diseases, infectious diseases, and endocrine diseases), and signal transduction (HIF-1 signaling pathway, PI3K-Akt signaling pathway, TNF signaling pathway, MAPK signaling pathway, and VEGF signaling pathway). Pathways unrelated to SLE were removed from the KEGG enrichment results, and a total of 28 enrichment results were obtained (Table 2). A targeted pathway network was constructed using Cytoscape 3.7.2, and the targeted areas and color tones in the network were arranged according to the Degree value. Among them, RELA, FOS, BCL2, PRCKA, CASP3, and CASP8 target genes rank at the top (Fig. 8). Each pathway interacts with common targets, indicating that Scutellaria baicalensis can treat SLE by coordinating multiple pathways.

Figure 7 Taxonomic analysis of KEGG enrichment pathways.

(A) Primary classification of KEGG pathway; (B) secondary classification of KEGG pathway.

Table 2 The top 28 pathways associated with SLE.

ID	Pathway	Count	P value	Genes	
hsa05417	Lipid and atherosclerosis	11	2.76E−10	CASP9, CYP2C9, IL6, CASP8, CASP3, BCL2, CYCS, PRKCA, PPARG, FOS, RELA	
hsa04933	AGE-RAGE signaling pathway in diabetic complications	7	3.93E−07	IL6, CCND1, CASP3, BCL2, PRKCA, RELA, VEGFA	
hsa04215	Apoptosis—multiple species	5	2.18E−06	CASP9, CASP8, CASP3, BCL2, CYCS	
hsa04115	p53 signaling pathway	6	2.29E−06	CASP9, CASP8, CCND1, CASP3, BCL2, CYCS	
hsa04210	Apoptosis	7	2.41E−06	CASP9, CASP8, CASP3, BCL2, CYCS, FOS, RELA	
hsa04066	HIF-1 signaling pathway	6	1.55E−05	IL6, BCL2, PRKCA, HIF1A, RELA, VEGFA	
hsa04917	Prolactin signaling pathway	5	5.15E−05	CCND1, FOS, ESR1, RELA, ESR2	
hsa05022	Pathways of neurodegeneration - multiple diseases	9	5.16E−05	CASP9, CHRM3, IL6, CASP8, CASP3, BCL2, CYCS, PRKCA, RELA	
hsa04151	PI3K-Akt signaling pathway	8	6.67E−05	CASP9, IL6, CCND1, BCL2, IGF2, PRKCA, RELA, VEGFA	
hsa04657	IL-17 signaling pathway	5	1.63E−04	IL6, CASP8, CASP3, FOS, RELA	
hsa01522	Endocrine resistance	5	1.92E−04	CCND1, BCL2, FOS, ESR1, ESR2	
hsa04659	Th17 cell differentiation	5	2.79E−04	IL6, FOS, AHR, HIF1A, RELA	
hsa04725	Cholinergic synapse	5	3.32E−04	ACHE, CHRM3, BCL2, PRKCA, FOS	
hsa04668	TNF signaling pathway	5	3.43E−04	IL6, CASP8, CASP3, FOS, RELA	
hsa04919	Thyroid hormone signaling pathway	5	4.30E−04	CASP9, CCND1, PRKCA, HIF1A, ESR1	
hsa04915	Estrogen signaling pathway	5	6.89E−04	BCL2, PGR, FOS, ESR1, ESR2	
hsa04623	Cytosolic DNA-sensing pathway	4	0.00141321	IL6, CASP8, CASP3, RELA	
hsa01521	EGFR tyrosine kinase inhibitor resistance	4	0.001641887	IL6, BCL2, PRKCA, VEGFA	
hsa04010	MAPK signaling pathway	6	0.001783353	CASP3, IGF2, PRKCA, FOS, RELA, VEGFA	
hsa04620	Toll-like receptor signaling pathway	4	0.004001105	IL6, CASP8, FOS, RELA	
hsa04726	Serotonergic synapse	4	0.004772032	CYP2C9, CASP3, PRKCA, PTGS1	
hsa04926	Relaxin signaling pathway	4	0.006569669	PRKCA, FOS, RELA, VEGFA	
hsa05418	Fluid shear stress and atherosclerosis	4	0.008071053	BCL2, FOS, RELA, VEGFA	
hsa04370	VEGF signaling pathway	3	0.013399869	CASP9, PRKCA, VEGFA	
hsa04621	NOD-like receptor signaling pathway	4	0.017715345	IL6, CASP8, BCL2, RELA	
hsa04510	Focal adhesion	4	0.022298783	CCND1, BCL2, PRKCA, VEGFA	

Figure 8 The Target-pathway network of the 28 enrichment results.

Molecular docking validation

Based on the above analysis, molecular docking was conducted between 5 core components (baicalin, acacetin, oroxylin A, neobaicalein, and stigmasterol) and 5 core targets (IL6, CASP3, BCL2, ESR1, and FOS). Affinity ≤ −4.25 kcal/mol indicates binding activity between the ligand and the target; affinity ≤ −5.0 kcal/mol indicates good binding activity; affinity ≤ −7.0 kcal/mol indicates strong docking activity. The best activities were selected in BCL2-stigmasterol (−8.1, Fig. 9A), IL-6-baicalein (−7.6, Fig. 9B), ESR1-baicalein (−7.6, Fig. 9C), and ESR1-acacetin (−7.5, Fig. 9D).

Figure 9 The results of molecular docking.

(A) Molecular docking results of IL-6-baicalein, (B) molecular docking results of BCL-2-stigmasterol, (C) molecular docking results of ESR1-baicalein (D).

Molecular dynamics simulation

To explore the stability of the protein-ligand interactions in greater depth, with a binding energy of ≤−7.6 kcal/mol as the limit, we selected the first four molecular docking systems for MD simulations: BCL2-stigmastero, IL-6-baicalein, ESR1-baicalein, and ESR1-acacetin, using GROMACS 2020.6 software. MD results such as RMSD, Rg value, and SASA provide an important basis for measuring the stability of complex systems of compounds and proteins, and the stability of protein tertiary structures after combining small molecules. As shown in Fig. 10A, the RMSD of all the complexes and proteins gradually stabilizes as the simulation progresses, indicating that the complexes are gradually stabilizing. The radius of gyration (Rg) can be used to describe the changes in the overall structure and is indicative of the compactness of protein structures, a larger change in Rg suggests that the system is more expanded. As depicted in Fig. 10B, the Rg values of the complexes remained stable throughout the simulation. The buried surface area (Buried SASA) of a small molecule embedded within a protein can reflect the size of the binding interface between the small molecule and the protein. As shown in Fig. 10C, the Buried SASA remains essentially stable, indicating that the contact area between the small molecule and the protein remains constant. The number of hydrogen bonds reflects the strength of protein-ligand binding, with ESR1-baicalein showing the highest hydrogen bond density and strength (Fig. 10D).

Figure 10 The results of molecular dynamic simulation.

(A) RMSD values of the four complexes. (B) Radius of gyration (Rg) values of the four complexes. (C) The evolution of the key distances between ligand features and the active site of the four complexes. (D) Number of hydrogen bonds in the four complexes.

The binding free energies (∆Gbind) of the four protein-ligand complexes were further calculated using the MM/GBSA method, three replicates per experiment, and the mean values + SD (E) are shown (Table 3). ΔEele represents the electrostatic interactions between the small molecule and the protein, ΔEvdw represents the van der Waals interactions, ΔEpol represents the polar solvation energy, which can indicate the electrostatic potential energy, and ΔEnonpol represents the nonpolar solvation energy, which can indicate the hydrophobic interactions. ΔEMMPBSA is the sum of ΔEele, ΔEvdw, ΔEpol, and ΔEnonpol, and the Gibbs Binding Energy (binding energy) ΔGbind is the sum of ΔEMMPBSA and −TΔS. As shown in Table 3, IL-6-Baicalein shows the lowest ∆Gbind value, indicating the strongest binding. Decompose the ΔEMMPBSA to obtain the contribution of each amino acid to the whole binding energy, the important residues were shown in Fig. S1A. We selected the top two residues and analysed the evolution of the distances between ligand features and the top two residues (Fig. S1B).

Table 3 The binding free energies of the four protein-ligand complexes.

Complex index	vdw	ele	pb	sa	mmpbsa (mm+pb+sa)	-TΔs	ΔGbind	
BCL2-stigmasterol	−88.709 ± 2.595	−31.689 ± 5.769	103.241 ± 4.877	−12.88 ± 0.028	−30.037 ± 3.334	12.629 ± 1.77	−17.408 ± 2.331	
IL-6-baicalein	−179.329 ± 0.714	−18.642 ± 0.753	70.96 ± 1.788	−23.503 ± 0.238	−150.514 ± 0.592	22.979 ± 1.6	−127.535 ± 1.022	
ESR1-baicalein	−149.214 ± 1.217	−58.762 ± 1.75	139.483 ± 3.669	−15.804 ± 0.112	−84.297 ± 1.746	15.32 ± 2.378	−68.977 ± 4.123	
ESR1-acacetin	−106.579 ± 0.557	−19.205 ± 1.109	109.301 ± 1.938	−13.024 ± 0.17	−29.506 ± 1.846	7.483 ± 0.749	−22.022 ± 1.503	
Note:

ΔEvdw, the van der Waals interactions; ΔEpol, the polar solvation energy; ΔEnonpol, the nonpolar solvation energy; ΔEMMPBSA = ΔEele +Δ Evdw + ΔEpol + ΔEnonpol, ΔGbind = ΔEMMPBSA − TΔS.

Discussion

SLE is a chronic autoimmune disease characterized by the loss of tolerance to self-antigens and the production of autoantibodies, which can lead to severe disability and death. Current treatment options for lupus are diverse but have issues such as poor specificity and significant adverse effects with long-term use. In recent years, an increasing number of studies have found that traditional Chinese medicine can significantly control the activity of SLE and reduce the dosage of corticosteroids (Wang et al., 2021a). Clinical studies have shown that the combination of Chinese herbal medicine with conventional Western medicine can effectively improve patient symptoms and shorten treatment time.

Analysis of potential active components

In this study, we preliminarily screened the effective components of Scutellaria baicalensis for treating SLE using network pharmacology, including baicalin, acacetin, quercetin, flavonoids of baicalin, and stigmasterol, which are also confirmed by high-performance liquid chromatography (HPLC) (Luo et al., 2024; Wang et al., 2025). Baicalin, a flavonoid compound extracted from the root of Scutellaria baicalensis, is the main active component of Scutellaria baicalensis and has been found in recent years to have good anti-tumor effects, including inhibiting cell proliferation, inducing cell apoptosis, autophagic cell death, and inhibiting cell adhesion, migration, and invasion. Animal experiments have shown that baicalin can improve renal function and pathological manifestations in lupus model mice by downregulating the activation of NLRP3 inflammasomes and the phosphorylation level of NF-κB (Li et al., 2019). Acacetin exhibits multiple effects, including antitumor, anti-inflammatory, and antiviral activities. Acacetin can inhibit the expression of Toll-like receptor 4 (TLR 4), tumor necrosis factor-related apoptosis-inducing ligand (TRAIL-R1), interleukin-6 (IL-6), tumor necrosis factor-alpha (TNF-α), and interleukin-5 (IL-5), thereby showing anti-inflammatory potential (Singh et al., 2020). In terms of anti-rheumatic disease, acacetin can inhibit the expression of matrix metalloproteinases in fibroblast-like synoviocytes through the MAPK signaling pathway, thereby playing a role in anti-rheumatoid arthritis (Chen et al., 2015).

Neobaicalein is a flavonoid compound isolated from Scutellaria baicalensis with significant anti-inflammatory effects. T helper cells 17 (Th17) and regulatory T (Treg) cells play an important role in the pathological process of autoimmune diseases. The imbalance of Th17/Treg cells is related to the occurrence and development of organ inflammation in systemic lupus erythematosus (Shan, Jin & Xu, 2020). In vitro experiments have confirmed that neobaicalin can inhibit the activation of STAT3 signaling by blocking the IL-17 signaling pathway, thereby improving the ratio of Th17 cells to Treg cells (Chen et al., 2022). It has been shown that stigmasterol can activate the NF-κB pathway through AMPK, thereby playing an anti-inflammatory role.

Targets analysis

This study indicates that the main targets regulated by Scutellaria baicalensis in SLE are IL-6, CASP3, BCL 2, ESR1, FOS, RELA, and PRCKA. IL-6 is related to T cell-dependent B cell activation and antibody production (Dienz et al., 2009), reduced natural killer (NK) cell cytotoxicity, and the amplification of pro-inflammatory cytokine cascades (Crayne et al., 2019). It has been reported that there is an increased B cell response to IL-6 in lupus patients (Kitani et al., 1992), and the inhibition of IL-6 levels in identified lupus patients helps to restore the balance of these naive B cells and T cells (Shirota et al., 2013). Anti-IL-6 treatment studies for SLE have been completed, in which the anti-IL-6 monoclonal antibody PF-04236921 was given to patients with active SLE and a placebo. The SLE response index-4 (SRI-4) and BICLA response rates were significantly different between the 10 mg treatment group and the placebo group in patients with higher disease activity (Wallace et al., 2017). Caspase-3 has a proteolytic role in programmed cell death (Nagata, 2018). A characteristic of T cells in SLE patients is an abnormal TCR-mediated signal transduction response, and it has been shown that casp3 can restore normal T cell function by limiting the key molecule CD3ξ of T cell receptor (Krishnan et al., 2005). FOS, as a universal immediate-early nuclear transcription factor, is involved in the regulation of cell proliferation, differentiation, transformation, and apoptosis. In animal experiments, FOS transgenic mice were more resistant to experimental autoimmune encephalomyelitis by inhibiting the production of inflammatory cytokines in dendritic cells (Wang et al., 2021b). The Fosl 1-JunB complex is related to the synthesis of IL-17 and the differentiation of Th17 cells. Their overexpression can lead to severe inflammatory symptoms and increased IL-17 expression. The RELA gene encodes the transcription factor p65, a subunit of NF-κB. The NF-κB signal is a key regulator of the immune system and inflammatory response. It has been shown that miR1013p can reduce the inflammatory response of peripheral blood mononuclear cells in SLE by inhibiting the expression of MAPK1 and blocking the NF-κB pathway (Zhao et al., 2021). PRCKA encodes a member of the serine/threonine-specific protein kinase family, protein kinase C (Coussens et al., 1986), which is involved in mediating cell growth and inflammatory responses (Cheng et al., 2020). PRKCA inhibits the release of pro-inflammatory cytokines through macrophages and the MAPK signaling pathway, thereby alleviating acute lung injury in lipopolysaccharide-induced mice (Wang et al., 2021c).

Biological enrichment analysis

GO analysis found that Scutellaria baicalensis plays a role in treating SLE by regulating oxidative stress, transcription-related, and apoptosis. KEGG signal pathway analysis indicates that the main signal pathways targeted by Scutellaria baicalensis intervention in SLE include the HIF-1 signaling pathway, PI3K-Akt signaling pathway, MAPK signaling pathway, IL-17 signaling pathway, which involves inhibiting inflammatory factors, affecting the expression and activation of immune cells, and regulating cell autophagy.

HIF-1 plays an important role in cellular hypoxia, inflammation, and tumorigenesis. In T cells, HIF-1 enhances the production of inflammatory cytokines and cytotoxic molecules. HIF-1α directly activates the expression of RORγt and inhibits the transcriptional activity of FoxP 3, regulating the development of Treg and Th17 cells (Dang et al., 2011). Overexpression of HIF-1α has been detected in human lupus CD4+ T cell subsets, indicating that abnormal expression of HIF-1α may be involved in the immune dysregulation of SLE (Garchow, Maque Acosta & Kiriakidou, 2021). Animal experiments have shown that increased expression of HIF-1 in CLE skin-infiltrating T cells promotes skin tissue damage, and inhibiting HIF-1 can eliminate disease progression (Little et al., 2023). The same changes have been observed in lupus nephritis mouse models (Chen et al., 2020). The PI3K-Akt signaling pathway can regulate a wide range of cellular processes, including cell proliferation, growth, and metabolism, especially in the nervous system. Excessive activation of the PI3K/AKT pathway in gliomas promotes the proliferation, invasion, and migration of glioma cells (Gao et al., 2022). The mitogen-activated protein kinase (MAPK) cascade signaling pathway is a key signaling pathway that regulates various cellular processes, including proliferation, differentiation, apoptosis, and stress responses. Animal experiments suggest that 1,25-dihydroxyvitamin D3 can significantly inhibit the NF-κB and MAPK signaling pathways to improve lupus activity, providing a potential target for the treatment of LN (Li et al., 2022).

Interleukin-17 (IL-17) is the signature cytokine of Th17 cells. The dysregulation of IL-17 expression is associated with a variety of inflammatory diseases, including inflammatory bowel disease (IBD), multiple sclerosis (MS), psoriasis, systemic lupus erythematosus (SLE), rheumatoid arthritis (RA), and asthma (Patel & Kuchroo, 2015). A study report showed that IL-17-dependent signaling in BXD 2 mouse cells leads to the production of autoantibodies and an increase in the number of germinal centers. In this study, IL-17-induced genes encode G protein signaling regulators 13 and 16 (Rgs 13 and Rgs 16), which inhibit the movement of B cells in the lymph node and promote the spontaneous formation of autoreactive germinal centers, thereby increasing the production of autoantibodies (Hsu et al., 2008). Significantly elevated levels of IL-17 can be detected in the plasma and serum of SLE patients (Crispin et al., 2008).

Molecular docking and dynamic simulation

Computational simulations were used to explore the binding affinity and mechanisms of Huangqin with key SLE-related targets. The molecular docking results show that the docking scores were all less than −5 kcal/mol. The best activities were selected in BCL2-stigmasterol, IL-6-baicalein, ESR1-baicalein, and ESR1-acacetin, primarily driven by hydrogen bonding and hydrophobic interactions. Subsequently, molecular dynamics simulations were conducted to analyze the dynamic behavior and stability of the complexes over time. The values of RMSD, Rg, and SASA and the number of hydrogen bonds indicated that the four complexes possessed reliable structural stability and compactness. Additionally, the binding free energies (∆Gbind) of the four protein–ligand complexes were analyzed by the MM/PBSA method, with the IL-6-baicalein showing the strongest binding affinities, which might inspire targeted drug delivery systems (Pingping, Nan & Yong, 2025) or derivative compounds optimized for efficacy and bioavailability (Gao et al., 2025).

Conclusions and limitations

In summary, this study used network pharmacology methods to explore the multi-target mechanism of Scutellaria baicalensis in treating systemic lupus erythematosus. The results show that the main signal pathways include the HIF-1 signaling pathway, PI3K-Akt signaling pathway, MAPK signaling pathway, and IL-17 signaling pathway. The mechanism involves inhibiting inflammatory factors, affecting the expression and activation of immune cells, and regulating cell autophagy. Our research may be instructive for the further development of effective components of Huangqin for the treatment of SLE.

Compared to the previous study about Huangqin treating SLE, we firstly analyse the molecular mechanism and the hub genes. In order to confirm the validity of hub genes, we phase out the pathways unrelated to SLE. Besides, with the help of molecular docking and dynamic simulation, we explain the interaction between Huangqin and SLE.

There are limitations about the study. Although we used molecular docking and dynamic simulation to mimic the combination and stabilization between the active components and targets, they can not clearly show the interaction between the compound and other proteins (inevitable in the actual environment). The postulated mechanism of action for the active components in traditional Chinese medicine requires further experimental verification using advanced techniques such as non-labeled molecular interaction systems. Therefore, further experimental validation are needed to be taken into account.

Supplemental Information

Supplemental Information 1 Information for potential targets of Huangqin components.

Supplemental Information 2 SLE-related 1638 potential targets in 3 databases.

Supplemental Information 3 PPI network of key targets.

Supplemental Information 4 BP, CC, and MF from GO enrichment analysis.

Supplemental Information 5 Signaling pathways from KEGG enrichment analysis.

Supplemental Information 6 The residues of the four complexes and the distance between the ligand and the protein.

(a) the residues of the four complexes (b) the evolution of the distances between ligand features and the top2 residues.

Supplemental Information 7 Raw data for dynamics simulation.

(A) Folder e1, e2, e3: the repetition files,

(B) File d_hbnum.xvg: the originnal file of hydrogen number,

(C) File d_rmsd_complex/ligand/protein.xvg: the originnal file of rmsd,

(D) File d_rmsf_portein.xvg: the originnal file of rmsf,

(E) File d_sasa_ complex/ligand/protein.xvg: the originnal file of buried sasa,

(F) File traj_0-100.pdb:the trajectory file,

(G) File complex_start/middle/end.pdb: the complex structure from initial to final.

Abbreviations

SLE Systemic lupus erythematosus

SLEDAI Systemic Lupus Erythematosus Disease Activity Index

TCMSP Traditional Chinese Medicine Systems Pharmacology Database and Analysis Platform

PPI Protein-Protein interaction

DAVID The Database for Annotation, Visualization and Integrated Discovery

GO enrichment analysis Gene Ontology enrichment analysis

BP biological process

MF molecular function

CC cellular component

KEGG enrichment analysis Kyoto Encyclopedia of Genes and Genomes enrichment analysis

PDB Protein Data Bank

MD Molecular dynamics

RMSD root-mean-square deviation

Rg radius of gyration

SASA solvent accessible surface area

Additional Information and Declarations

Competing Interests

The authors declare that they have no competing interests.

Author Contributions

Hui Yang conceived and designed the experiments, performed the experiments, analyzed the data, authored or reviewed drafts of the article, and approved the final draft.

Shuting Zheng conceived and designed the experiments, performed the experiments, analyzed the data, prepared figures and/or tables, authored or reviewed drafts of the article, and approved the final draft.

Jialing Wu conceived and designed the experiments, authored or reviewed drafts of the article, and approved the final draft.

Ou Jin conceived and designed the experiments, authored or reviewed drafts of the article, and approved the final draft.

Xi Zhang conceived and designed the experiments, authored or reviewed drafts of the article, and approved the final draft.

Data Availability

The following information was supplied regarding data availability:

The sequences are available at GEO: GSE51997. The GeneCards database (https://www.genecards.org/) was searched with “[disorders](systemic lupus erythematosus)” and the KEGG database (https://www.kegg.jp/) was searched for “Systemic lupus erythematosus”.

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
