# Peer review of "The potential mechanism of Huangqin for treatment of systemic lupus erythematosus based on network pharmacology, molecular docking and molecular dynamics simulation"

_PeerJ, doi:10.7717/peerj.19536_

## Round 0.1 · original submission · Major Revisions

Three experts in the field have reviewed your manuscript. Although all admitted the potential value of this work, they requested extensive revisions. Please read their comments carefully and revise the manuscript accordingly. I believe those by Reviewer 3 are particularly important among the reviews.

Reviewer 1 ·

Basic reporting

The potential mechanism of Huangqin for treatment of systemic lupus erythematosus based on network pharmacology, molecular docking and molecular dynamics simulation
Zheng et al. studied the active ingredients of TCM Huangqin for treating SLE.
Main comments:
1. The authors claimed that Huangqin can treat SLE through its effective components such as baicalin and acacetin. However, these results are not experimentally validated.
2. There is little confirmatory assays on the effect of Huangqin on SLE, for example, is Huangqin truly consisting of baicalin and acacetin?

Experimental design

see above

Validity of the findings

see above

Additional comments

see above

·

Basic reporting

No comment

Experimental design

1.The last paragraph (4th paragraph) of the introduction has sufficiently explained how this study builds upon prior research on traditional Chinese medicine, particularly in relation to SLE. However, it could be further strengthened by explicitly addressing the knowledge gaps that remain unanswered in previous studies regarding the use of baicalin or the treatment of SLE with traditional Chinese medicine.

2.Expand the discussion on the limitations of computational predictions and the need for experimental validation to support findings.

Validity of the findings

1. Explicitly address how this study advances the field beyond previous findings. For example, compare your results with prior studies on Huangqin or similar herbal treatments for autoimmune diseases.

Additional comments

The manuscript is comprehensive and methodologically robust, combining advanced computational tools. However, consider expanding the discussion to include potential clinical implications of targeting the identified pathways for SLE management.
for example:
Drug Development, the molecular insights provided in this study could inform the development of novel drug formulations. For instance, baicalin’s demonstrated binding stability with IL6 might inspire targeted drug delivery systems or derivative compounds optimized for efficacy and bioavailability.

·

Basic reporting

The quality of the English in the document is acceptable. However, the wording needs to be revised. The references are appropriate for the main concepts discussed in the document. The relevance of the data in the form of tables or figures should be reviewed, and consideration should be given to improving their presentation. A well-presented data can significantly enhance the impact of the work. Ensuring consistency of content and visibility is also crucial. Since both figures and tables lack figure captions, they should be integrated, and the description of the main abbreviations used should be considered.

Experimental design

The research questions are relevant to molecular rheumatology. The methods are well described; however, about the analysis platform with which the search for the physicochemical properties of Scutellaria baicalensis was performed, is there information on any other database to perform the search? A revision of the wording of certain statements is necessary. For instance: on lines 147 to 148 is the following sentence: "The darker the color of the nodes in the diagram, the greater the impact on SLE".

Validity of the findings

The results obtained with molecular docking demonstrate the significant potential of Huangqin's active compounds to influence the inhibition of some signaling pathways that may be related to the pathology of SLE. This potential is a key focus of our research. However, it is suggested to review the abbreviations used and not described from the introduction to the discussion section since a large number of them are used throughout the document, causing confusion and misunderstandings.

---

## Round 0.2 · Minor Revisions

This time, only Reviewer 3 sent me back the review comments. As you can confirm, the reviewer still requests another round of revisions of the writing and syntax of the manuscript. One of the PeerJ staff members has the same opinion:

There are several minor grammatical issues throughout the manuscript. For example:

In the abstract: "based on network pharmacology , and further analyze the potential mechanisms of action."

There is an unnecessary space before the comma, and the phrase "analyze and explore" is a bit repetitive and could be reworded for clarity.

There are numerous instances where the spacing between words or sections is inconsistent or missing. For example:

"Results:Network pharmacology..." - Missing space after colon.
"Stigmastero, Baicalein and Acacetin.The dynamics simulation..." - Missing space after period.

Therefore, I decided to request minor revisions. Please review your manuscript one more time and polish it as much as possible.

**Language Note:** The Academic Editor has identified that the English language must be improved. PeerJ can provide language editing services - please contact us at [email protected] for pricing (be sure to provide your manuscript number and title). Alternatively, you should make your own arrangements to improve the language quality and provide details in your response letter. – PeerJ Staff

·

Basic reporting

The authors complemented the article with suggestions to clarify the main idea of the article.

Experimental design

It is considered that the suggestions and changes requested to improve the understanding of the methodology section were integrated.

Validity of the findings

Pertinent limitations were added to the study.

Additional comments

It is recommended that the manuscript be completely revised in writing and syntax, including the separation of sections and words.

---

## Round 0.3 · accepted · Accept

Since the requested points were addressed, I am happy to accept the manuscript (though I am not confident that there are no more grammatical issues).